# Mitigating Catastrophic Forgetting in Spiking Networks through Threshold Modulation

**Ilyass Hammouamri**                                    *ilyass.hammouamri@cnrs.fr*
*CerCo CNRS UMR 5549, Université Toulouse III, Toulouse, France*

**Timothée Masquelier**                                  *timothee.masquelier@cnrs.fr*
*CerCo CNRS UMR 5549, Université Toulouse III, Toulouse, France*

**Dennis Wilson**                                        *dennis.wilson@isae.fr*
*ISAE-Supaero, Université de Toulouse, Toulouse, France*

**Reviewed on OpenReview:** *https://openreview.net/forum?id=15SoThZmtU*

## Abstract

Artificial Neural Networks (ANNs) trained with Backpropagation and Stochastic Gradient Descent (SGD) suffer from the problem of Catastrophic Forgetting; when learning tasks sequentially, the ANN tends to abruptly forget previous knowledge upon being trained on a new task. On the other hand, biological neural networks do not suffer from this problem. Spiking Neural Networks (SNNs) are a class of Neural Networks that are closer to biological networks than ANNs and their intrinsic properties inspired from biology could alleviate the problem of Catastrophic Forgetting. In this paper, we investigate if the firing threshold mechanism of SNNs can be used to gate the activity of the network in order to reduce catastrophic forgetting. To this end, we evolve a Neuromodulatory Network that adapts the thresholds of an SNN depending on the spiking activity of the previous layer. Our experiments on different datasets show that the neurmodulated SNN can mitigate forgetting significantly with respect to a fixed threshold SNN. We also show that the evolved Neuromodulatory Network can generalize to multiple new scenarios and analyze its behavior.

## 1 Introduction

Agents in nature have the ability to learn novel tasks while still retaining previously acquired knowledge, it is a fundamental quality of biological neural networks. In contrast, Artificial Neural Networks (ANNs) suffer from the infamous problem of Catastrophic Forgetting (CF) (McCloskey & Cohen, 1989; French, 1999), where learning a new task causes the ANN to completely and abruptly forget previously learned tasks. Usually, this problem is not encountered when the the data is shuffled aka independent and identically distributed (i.i.d.). Most, if not all, of deep learning's (DL) achievements are in the i.i.d. case. However, a large portion of problems in the real world requires agents to learn sequentially from new encountered data.

Spiking Neural Networks (SNNs) are a class of neural networks which are more biologically inspired than standard ANNs (Maass, 1997). In SNNs, neurons are stateful and emit a spike only when their potential exceeds a certain threshold. Recently, it has been made possible to train deep SNNs using backpropagation and gradient descent through surrogate gradients (Neftci et al., 2019; Zenke et al., 2021) to achieve good performance on classification tasks. Furthermore, SNNs make an interesting candidate for many real world applications due to their properties such as energy efficiency and fast inference when implemented on neuromorphic hardware. However, the field of SNNs is still in its early phases, and their properties are still under study.

When learning sequentially using an ANN, training on a new task with backpropagation and SGD will continue to modify the weights of the network, including weights that were important for previous tasks,

and push them in the direction of the new task. This is not the case in SNNs, as weight update only occurs when the corresponding pre-synaptic neuron's membrane potential reaches a threshold and emits a spike, unlike ANNs where the weights of a neuron with a very small activation value still gets updated. This mechanism could potentially serve as a task-dependent gating, where weights that are important for previous tasks are preserved if the corresponding neurons do not fire during the learning of new tasks. Previous works have explored artificially reproducing this mechanism in ANNs: Masse et al. (2018) used a gating signal which informs the network explicitly of the task it is being trained or tested on and Beaulieu et al. (2020) used a neuromodulatory network that produces an element-wise multiplicative mask to gate the activations.

In this paper, we investigate if the intrinsic mechanisms of SNNs can be leveraged to gate neural activity in a way that mitigates catastrophic forgetting. To achieve this, we use a neuromodulatory network that adapts the firing threshold of neurons. This neurmodulatory network only has access to the local information of the previous layer's spiking activity, and not the network input data as in Beaulieu et al. (2020). Instead of introducing an artificial gating mechanism, we use the inherent gating of SNNs and take direct inspiration from neuromodulatory mechanisms in the brain by acting on the threshold of spiking neurons.

## 2 Background

### 2.1 Neuron Models

The neuron models we use for our SNNs are The Leaky Integrate-and-Fire (LIF) model (Gerstner et al., 2014) for dynamic datasets and the Integrate-and-Fire (IF) model (without leak) for static datasets. The LIF model is widely used in computational neuroscience and machine learning for its simplicity and performance.

The sub-threshold dynamics of the $i^{th}$ LIF/IF neuron are described by the following differential equation:

$$\tau \frac{dV^i}{dt} = f(V^i(t), I^i(t)) \tag{1}$$

where $f$ is a function that depends on the neuron's model, for the LIF and IF, $f$ is described by Eq.2 and Eq.3 respectively.

$$f(V^i(t), I^i(t)) = -(V^i(t) - V_{rest}) + I^i(t) \tag{2}$$

$$f(V^i(t), I^i(t)) = I^i(t) \tag{3}$$

In the equations above, $V^i$ is the neuron's membrane potential, $I^i$ is the input current received by the neuron, $\tau$ is the membrane time constant and $V_{rest}$ is the neuron's potential at rest.

For numerical simulation, following Fang et al. (2021), we use discrete-time equations that are an approximation of the differential equations above:

$$H^i[t] = f(V^i[t-1], I^i[t]) + V^i[t-1] \tag{4}$$

$$V^i[t] = H^i[t](1 - S^i[t]) + V_{rest}S^i[t] \tag{5}$$

$$S^i[t] = \Theta(H^i[t] - V_{th}) \tag{6}$$

where $H^i[t]$ and $V^i[t]$ are the membrane potential of the $i^{th}$ neuron at time-step $t$, after the integration of the input current $I^i[t]$ (Eq. 4) and after the spiking dynamics (Eq. 5 and 6), respectively. $\Theta$ is the Heaviside function; $S^i[t]$ equals 1 if the membrane potential is above the firing threshold $V_{th}$ (neuron $i$ fires at time-step $t$) and 0 otherwise. If a spike occurs, the membrane potential is instantly reset to $V_{rest}$, which corresponds to a so-called "hard reset", otherwise it follows the sub-threshold dynamics (Eq.5). As suggested by Zenke & Vogels (2021), we ignore the neuronal reset when computing gradients by detaching them from the computational graph.

The input current $I^i[t]$ is defined as

$$I^i[t] = \sum_j w_{ij} S^j[t] \tag{7}$$

where $w_{ij}$ denotes the synaptic weight between neuron $i$ and the afferent neuron $j$.

If a neuron $i$ is subject to neuromodulation, its firing threshold $V_{th}^i[t]$ at time $t$ is not constant, but instead it is a function of time and the network's spiking activity:

$$V_{th}^i = g(t, S[t]). \tag{8}$$

## 2.2   Surrogate Gradient Learning

Recently, it has been made possible to train deep SNNs in a supervised manner using backpropagation and gradient descent as in ANNs through surrogate gradient learning (SGL) (Neftci et al., 2019; Zenke et al., 2021). One of the key challenges in this method is the non-differentiability of the spiking function. The derivative of Eq.6 is required by backpropagation, however, the Heaviside function's derivative is zero almost everywhere and not defined at zero, which makes LIF neurons unsuitable for gradient based optimization.

To overcome this issue, Neftci et al. (2019) proposed using a surrogate gradient as a smooth continuous relaxation to the spiking function. During the forward pass, the spiking function is unchanged, whereas in the backward pass the derivative of the Heaviside function is approximated by the derivative of a smooth continuous function. In our work, we use the sigmoid function for static datasets.

$$\Theta'(x) \approx \sigma'(x) = \sigma(x)(1 - \sigma(x)) = \frac{e^{-x}}{(1 + e^{-x})^2} \tag{9}$$

and atan (arc tangent) function for the dynamic dataset

$$\Theta'(x) \approx \arctan'(x) = \frac{1}{1 + x^2} \tag{10}$$

## 2.3   Continual Learning

Continual Learning concerns the problem in which a single neural network must learn the tasks $\mathcal{T} = (\mathcal{T}_1, \mathcal{T}_2, ..., \mathcal{T}_n)$ in a sequential manner and under the condition that only the data of the current task is available for training, without catastrophically forgetting the previously learned tasks. This differs from standard learning in ANNs, where data from multiple tasks $\mathcal{T}$ are presented in mixed mini-batches sampled independently and which are meant to be identically distributed (i.i.d.).

However, due to differences in experimental protocols, the difficulty of the problem can change drastically. A categorization scheme is proposed in van de Ven & Tolias (2019), where different continual learning scenarios are defined based on the availability of the task-ID to the network during training. Task-IL, domain-IL and Class-IL (in order increasing difficulty, where IL stands for Incremental Learning) denote the scenarios where the task-ID is provided, task-ID is not provided and task-ID needs to be inferred, respectively. Another categorization scheme proposed by Farquhar & Gal (2018) is based on the architecture of the network to distinguish between multi-headed and single-headed architectures. A multi-headed architecture has separate output layer parameters for each task (and requires the task-ID to switch between heads), while in the single-headed architecture, all the parameters are shared between tasks.

Multiple solutions have been proposed to alleviate the problem of catastrophic forgetting, such as replaying certain data, adding new parameters, adjusting learning through meta-learning, and finally selective learning of a subset of network parameters.

Replay methods (Rolnick et al., 2019) rely on storing previously encountered data or a representation of it in order to be combined with new data which leads to an approximation of an i.i.d. sampling over all of the sequentially encountered data. In these methods, the learning system is retrained over previous data many times. While replay methods can greatly alleviate catastrophic forgetting, they do not scale well with the number of sequential tasks.

Similarly, other solutions such as Rusu et al. (2016) avoids interference between tasks by adding new separate parameters for each new task. These new parameters are trained exclusively on the new task, inheriting information from layers in previous tasks with frozen weights. While this approach allows for high performance in the different task sub-networks, it similarly suffers from poor scaling with the number of tasks.

Several works use meta-learning (Finn et al., 2017) to adjust the learning process for continual learning (Javed & White, 2019; Xu et al., 2020; Harrison et al., 2020). Meta-learning aims to optimize the learning of a model over a distribution of learning tasks such that learning generalizes to other tasks, usually through few-shot learning on a test set of tasks. Such approaches can therefore naturally be expanded to continuous learning, as in Javed & White (2019) which learns high-level representations which aim to minimize catastrophic interference between tasks.

The approaches which are closest to this work use selective plasticity to control the modification of parameters during weight update. In Kirkpatrick et al. (2017) and Zenke et al. (2017) the modification of parameters during training is controlled by a regularization term that penalizes changes to important parameters. In Masse et al. (2018) and Beaulieu et al. (2020), a gating signal is fed to the network in order to block some neurons from being activated or mask certain parameters to prevent them from being updated during new tasks. Our method uses threshold modulation to control which parts of the network can spike more or less easily which then influences weight update.

The problem of catastrophic forgetting specifically in SNNs has also been studied. Vaila et al. (2020) proposes a regularization term that penalizes the update of certain parameters termed the cost per synapse as a metric of the importance of every parameter in respect to previous tasks. In Vaila et al. (2020), a part of the feature extraction section of the network is spiking and was pretrained using STDP; this spiking part is then frozen and only the non-spiking classification layers are subject to regularization. Allred & Roy (2020) proposes a single-layer SNN trained in a unsupervised manner using STDP for continual learning on MNIST using dopaminergic excitatory neurons that stimulate other neurons to fire. Our work differs from these works in the use of a convolutional SNN trained with backpropagation regulated through spiking threshold modulation.

## 3 Methods

### 3.1 Task

The scenario of Continual Learning we adopt in this study is single-headed Class Incremental-Learning (van de Ven & Tolias, 2019; Farquhar & Gal, 2018). The model is neither informed of the task ID nor of the occurrence of a change in tasks and all of the parameters of the model are shared between tasks. This is considered the most difficult Continual Learning scenario in van de Ven & Tolias (2019)

In the experiments presented in this paper, a task $\mathcal{T}_i$ consists of learning to correctly classify a single class $C_i$ from a dataset $\mathcal{D}$. More precisely, in step $i$ the model is trained using a small number $k$ of instances of class $C_i$. Seeing that all the parameters (including the output layer) are shared, the challenge is to learn to classify the current class $C_i$ correctly with few examples without forgetting the previously learned classes $(C_j)_{j<i}$.

To achieve this, we modulate the spiking thresholds of the final fully connected (FC) layer of our model using an external network that we call the Neuromodulator Network (NmN), that has been evolved to mitigate catastrophic forgetting. SNNs equipped with an NmN are referred to as Neuromodulated-SNN (Nm-SNN) and standard SNNs without neuromodulation are referred to as SNN.

Each dataset $\mathcal{D}$ is divided into three partitions with different classes: classes in $\mathcal{D}_{pre}$ are used for pre-training, classes in $\mathcal{D}_{evo}$ are used in the evolutionary optimization of the NmN and classes in $\mathcal{D}_{test}$ are used to test the generalization of our method. Moreover, each dataset $\mathcal{D}$ is divided into 80% training instances and 20% testing instances.

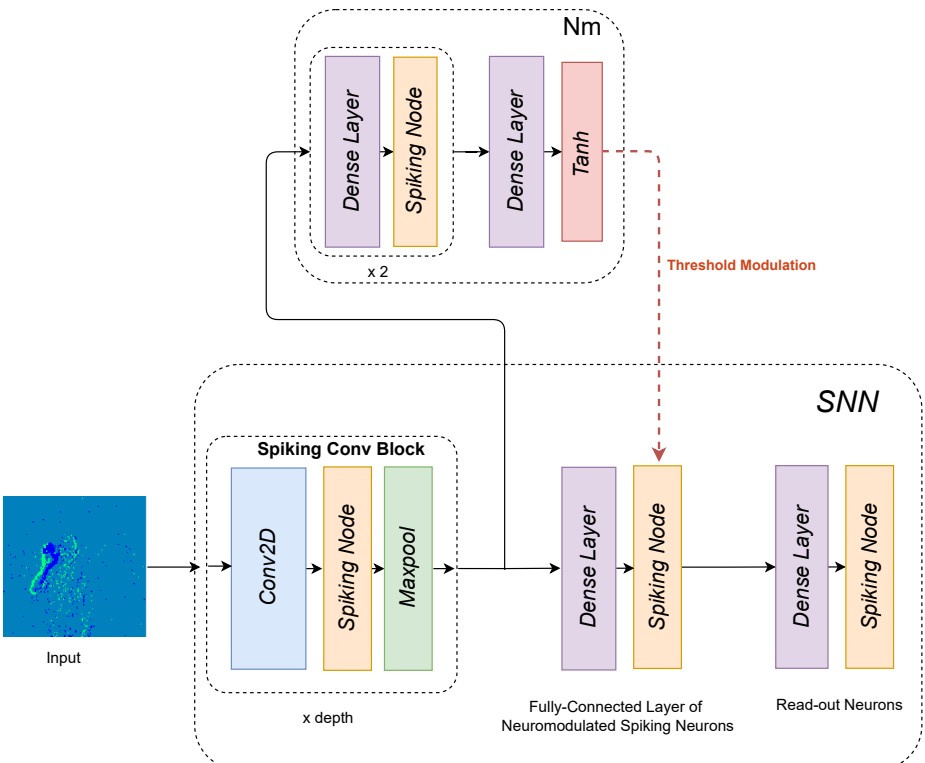

Figure 1: The Nm-SNN architecture is composed of two spiking networks: a SNN and the NmN. The convolutional SNN performs classification; the first convolutional blocks and the subsequent linear weights are pre-trained on $\mathcal{D}_{pre}$ using SGL and frozen during the CL scenario. The fully connected spiking network (NmN) takes as input the spiking activity of the final SNN convolutional layer and outputs the threshold adaptation value for each spiking neuron in the final hidden layer of the SNN during each time-step.

## 3.2 Neuromodulated SNN

We present the Nm-SNN's architecture (Figure 1), which consists of two separate networks, a convolutional SNN that learns the classification task and the Neuromodulator Network (NmN) which is responsible for adapting the spiking thresholds of the final fully connected (FC) layer of the SNN. We chose to only modulate the last layer as the first layers have been shown to have only a small contribution to the observed forgetting in neural networks (Ramasesh et al., 2021).

The convolutional SNN, as shown in Figure 1, is composed of a spiking convolutional block with a depth that depends on the complexity of the task $\mathcal{T}$ which performs a two dimensional convolution operation on the input with an activation using a LIF model for dynamic datasets or IF model for static datasets. This is followed by a Max Pooling layer and a fully connected layer (FC) of $n_{fc}$ LIF (or IF) neurons with adaptive thresholds. Finally, the output layer has $n_{classes}$ spiking neurons.

We use a direct input encoding; the first layer neuron's input currents correspond to the pixel intensities of the current frame $x[t]$ (for static datasets, $x[t] = x$ is constant throughout the simulation time $T$). The prediction of the network is chosen based on the spike count over the whole simulation time $T$ in the output layer where each output neuron corresponds to a certain predefined class.

The SNN's convolutional block and its subsequent fully connected layer are pre-trained using SGL on classes from $\mathcal{D}_{pre}$ and are frozen during the CL scenario. We found that this pre-training step can improve performance significantly, seeing that we use a very small number of examples for each class. The classes in $\mathcal{D}_{pre}$ are neither used in the evolutionary optimization nor in the testing phase.

The SNN is equipped with a threshold Neuromodulator Network (NmN) which is a simple Fully connected Spiking Neural Network with two hidden FC layers of LIF neurons. The NmN interacts with the SNN by modulating the spiking thresholds of its final FC, both in the training and inference phases (see Figure 1 ). More precisely, at each time-step $t$, the NmN takes as input $S_{conv}[t]$ the spikes emitted from the convolutional block at time-step $t$ and outputs a vector $\Delta V_{th}[t]$ that has the same size as the SNN's FC layer, with values in $[-1, 1]$ that corresponds the change to apply to the threshold of each neuron in the FC layer. We don't allow threshold values to be zero or negative (for biological plausibility purposes), thus the final threshold adaptation is $max(0, V_{th}[t] + \Delta V_{th}[t]) + \epsilon$ . After $T$ time-steps (the total simulation time for an input instance), all the spiking thresholds are reset to a base value $V_{th,0}$ (see Algorithm 1 [1]).

During each task of the CL scenario, the SNN's trainable parameters are optimized using SGL, while the NmN's parameters are frozen and are optimized after the end of the CL scenario using an Evolutionary Strategy detailed in subsection 3.3.

---

**Algorithm 1** : Neuromodulated training step

---

**Require:** $SNN, NmN, T$ : Total timesteps,
  $x$ : Input, $y$ : Label, $h$ : Predictions

  Reset SNN and NmN tresholds
  **for** $t = 1, 2, ..., T$ **do**
      $Sconv[t] \leftarrow SNN.conv\_block(x[t])$                    ▷ Get the conv block's spiking activity
      $\Delta V_{th}[t] \leftarrow NmN(Sconv[t])$                   ▷ The adaptation value from the NmN
      $SNN.FC.V_{th}[t] \leftarrow SNN.FC.V_{th}[t] + \Delta V_{th}[t]$     ▷ Adapt the SNN's FC thresholds
      $h[t] \leftarrow SNN.FC(Sconv[t])$
  **end for**
  $loss \leftarrow calculate\_loss(h, y)$
  Update SNN's weights using $loss$

---

### 3.3 Neuromodulator Optimization

In order to optimize the NmN, we use a canonical $(\mu/\lambda)$ Evolutionary Strategy $((\mu/\lambda) - ES)$ (Chrabąszcz et al., 2018), where $\mu$ and $\lambda$ are the parent population size and the offspring population size, respectively. The population is composed of vectors that correspond to the parameters of NmN. Individuals are generated by adding random noise from $\mathcal{N}(0, \sigma^2)$ to a distribution center and then updating the center based on fitness information from the individuals.

Another approach to optimize the NmN would be to run a CL scenario (i.e. learn $n$ sequential tasks containing $k$ examples, with the Nm-SNN) and then backpropagate the error using the surrogate gradient through a computational graph constructed over all the sequential tasks. This is similar to the approach taken in Beaulieu et al. (2020) on ANNs. However, we found this approach to have significant memory requirements and could only use it for very small $n$ and $k$ as we calculated gradients over the entire CL scenario. An evolutionary method allows for optimization of the NmN parameters without the gradient signal at the cost of longer total computation time (each generation takes a considerable amount of time to compute) and with the critical choice of the fitness function that we optimize for.

We chose a fitness function (Algorithm 2) that encompasses the entirety of the CL scenario, by taking the weighted mean of the classification accuracies $acc_{ij}$ over all learned tasks (including the current one) at every step of the CL scenario. $acc_{ij}$ denotes the classification accuracy of task $j$ at the CL step $i$ (where $i \geq j$ ), thus we take into consideration the accuracy of remembering previous tasks $j < i$ as well as the accuracy of learning the current task $i$ at every step of the CL scenario. We evaluate the fitness of each NmN in the population by equipping it to the SNN and carrying out the CL scenario for different task permutations. In Algorithm 2, $T_{ip}$ denotes the $i^{th}$ task in permutation $p$, $acc_{ijp}$ denotes the classification accuracy of task $j$ evaluated at step $i$ of the CL scenario with permutation $p$.

---

[1]Code available at https://github.com/Thvnvtos/Nm-SNN

---

**Algorithm 2** : Fitness function

---

**Require:** $SNN, NmN, \mathcal{T}$ : Tasks, $P$ : Set of permutations
 $\alpha$ : Positional coefficients

 $SNN = reinitialize\_SNN()$        ▷ reinitialize neuron's potentials and thresholds
 $SNN.nmn = NmN$        ▷ equip current NmN to SNN
 **for** $p$ in $P$ **do**        ▷ iterate through all task ordering permutations
     **for** $i = 1, 2, ..., n$ **do**        ▷ $n$ is the number of tasks in $\mathcal{T}$
        **for** $x$ in $T_{ip}$ **do**
           $SNN \leftarrow train(SNN, Nm, x)$        ▷ Algorithm 1
        **end for**
        **for** $j = 1, 2, ..., i$ **do**
           **for** $x$ in $T_{jp}$ **do**
              $f \leftarrow evaluate(SNN, Nm, x)$        ▷ evaluate on previous tasks (Algorithm 1)
              $acc_{ijp} \leftarrow \alpha_{ij} f$
           **end for**
        **end for**
     **end for**
 **end for**
 **return** $Mean(\boldsymbol{acc})$

---

We observed that the order at which the sequential tasks are learned by the network has a significant effect on the severity of catastrophic forgetting; it is possible to find a task ordering where the interference between different tasks is minimal and thus allows for an easier retention of information. Furthermore, the performance of an Nm-SNN optimized on a single specific task ordering does not necessarily generalize to different orderings. For this reason, our fitness function evaluates an individual on multiple permutations of the task ordering: for $n$ tasks, we use $n$ permutations obtained by a translation to the right.

Finally, we use positional coefficients $\alpha$ to calculate the weighted mean of accuracies to compensate for the differences in the difficulties between tasks. For example, at CL step $i$, remembering the immediate previous task $i - 1$ is significantly easier than remembering the first learned task 1.

## 4 Results

Our goal was to compare the performance of a standard SNN and a Nm-SNN, where both networks have the same architecture, same parameter size, same pre-trained frozen weights and the same trainable parameter initialization. The only difference lies in the spiking thresholds of the final FC layer. For the standard SNN, the spiking thresholds are constant with $V_{th} = 1$ for all neurons, while in the case of the Nm-SNN, the spiking thresholds are adaptive and are modulated by the NmN.

To this end, we use two different type of datasets: a neuromorphic dataset DVS128 Gesture (Amir et al., 2017) composed of hand gestures captured with an event-based camera, and static image datasets EMNIST (Extended-MNIST) letters (Cohen et al., 2017) composed of images of handwritten uppercase and lowercase letters and MNIST (Deng, 2012). For all experiments, the tasks (as defined in subsection 3.1) consist of learning to correctly classify a single class. We used SpikingJelly (Fang et al., 2020) which is a PyTorch-based open-source deep learning framework for SNNs.

We also compare with an SNN trained using EWC (Kirkpatrick et al., 2017), a state-of-the-art method for continual learning, which we term EWC-SNN. The experimental protocol for using EWC differs from the training method used for the neuromodulated and standard SNNs. When switching between tasks, EWC recomputes gradient information from the previous task to inform weight updates on the next task, requiring knowledge of task change. The neuromodulated and standard SNNs are trained over multiple tasks completely online without any information transfer between tasks or about task change. EWC-SNN also has the same architecture, parameter size and pre-trained frozen weights as the other models, on each task

update we calculate the approximation of the fisher information term using all data from the previous task, and for each of the settings presented below, we use a simple search method to find the best $\lambda$ value, the coefficient for the EWC regularization term.

We validated our method by three tests of generalization. First, we test the generalization to new instances that were not used during the evolutionary algorithm from the tasks in $\mathcal{D}_{evo}$. Second, we test the generalization to new tasks $\mathcal{D}_{test}$, unseen during both the evolution and pre-training. Third, we test the generalization to a bigger number of sequential tasks $n$ than the one used for the evolutionary optimization. For each of these settings, we also test the generalization to a larger number of gradient steps than the one used during evolution.

For each setting, we also test on $n$ different permutation of the tasks ordering (as defined in subsection 3.3). The SNN is reinitialized for every CL scenario and is optimized using SGL and the neurmodulator is the evolved NmN with the best fitness during evolution (with frozen weights).

## 4.1 DVS128 Gesture

For DVS128 Gesture, the evolution configuration is as follows: we use the classes from $\mathcal{D}_{evo}$ with $n = 3$ sequential tasks, each class is learned through 20 SGD updates where we have $k = 40$ instances of each class and a batch size of 2, each instance consists of $T = 16$ frames (after pre-processing of the event data) which is also our simulation time. The evolution of the NmN lasted approximately 600 generations until convergence.

As stated before, for the $\mathcal{D}_{evo}$ setting we use new instances of the classes that were not used during the evolutionary strategy, except for the 50 gradient step setting in DVS128 Gesture, where part of the EA instances were reused since the total number of examples of each class in the dataset is limited.

For the third test, due to having only 11 different classes, we mix $\mathcal{D}_{evo}$ and $\mathcal{D}_{test}$ (we alternate tasks from each dataset instead of concatenation for more difficulty) to obtain $n = 6$ sequential tasks, we note this dataset as $\mathcal{D}_6$. The concatenation was done as follows. If $\mathcal{D}_{evo} = \{a, b, c\}$ and $\mathcal{D}_{test} = \{e, f, g\}$ then $\mathcal{D}_6 = \{a, e, b, f, c, g\}$.

Table 1: Mean accuracies for DVS128 Gesture

| Setting | SNN (20) | EWC-SNN (20) | Nm-SNN (20) | SNN (50) | EWC-SNN (50) | Nm-SNN (50) |
|---------|----------|--------------|-------------|----------|--------------|-------------|
| $\mathcal{D}_{evo}$ | 66.89% | 83.56% | **98.09%** | 61.98% | 72.69% | **85.19%** |
| $\mathcal{D}_{test}$ | 75.35% | 85.88% | **99.02%** | 64.87% | 82.18% | **91.72%** |
| $\mathcal{D}_6$ | 45.32% | 51.06% | **64.65%** | 39.89% | 44.41% | **56.37%** |

Table 1 shows that the Nm-SNN significantly outperforms the standard SNN in terms of mean accuracy and successfully generalizes to new settings. Nm-SNN also outperforms EWC-SNN on both settings and across datasets. The mean is taken over all accuracies at every stage of the sequential learning (previous classes accuracies are re-calculated whenever a new class is learned, and not just at the end) and all $n$ permutations and different random seeds. We note for the hardest setting of $\mathcal{D}_6$ (50), with $n = 6$ sequential classes and 50 SGD updates per class, there is a sharp drop in mean accuracy due to forgetting, however the performance improvement between the Nm-SNN and standard SNN is still similar to the easier settings. Finally, the performances on $\mathcal{D}_{test}$ are better than $\mathcal{D}_{evo}$ for Nm-SNN, even though it has been optimized for $\mathcal{D}_{evo}$. This might be due to the fact that the classes of $\mathcal{D}_{test}$ have less interference between them as seen in the imporvement of the standard SNN as well.

Figure 2 allows for a more thorough comparison of the Nm-SNN and SNN accuracy distributions. The figure shows that the mean accuracy for paired samples in both settings is significantly higher for Nm-SNN, the paired t_test value for $\mathcal{D}_{test}$ (20) is 13.751 (with $p = 2.79 \times 10^{-10}$ ) and 7.12 (with a $p = 2.43 \times 10^{-06}$ ) for $\mathcal{D}_{test}$ (50). A remarkable difference between the (20) and (50) distributions is that the interquartile range is much larger in the (50). We presume that this is due to the fact that some permutations become significantly harder as the number of SGD updates grows.

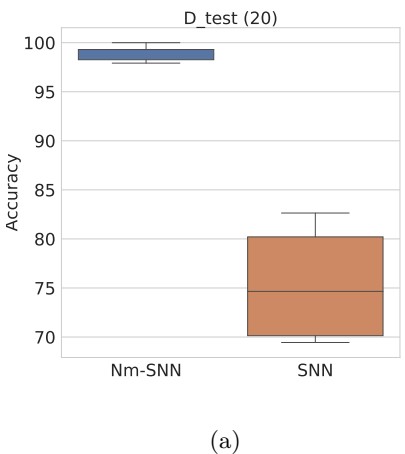 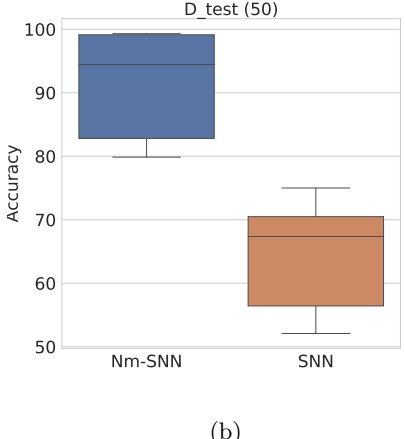

|     |     |
| --- | --- |
| (a) | (b) |

Figure 2: Distribution of accuracies of Nm-SNN (in blue) and SNN (in orange), for the settings of $\mathcal{D}_{test}$ (20) Figure 2a and $\mathcal{D}_{test}$ (50) Figure 2b for DVS128 Gesture Dataset

As we previously mentioned in subsection 3.3, the order in which the classes are learned is crucial to the performance of the network in terms of retaining previous information; some permutations are favorable to continual learning in such as learning the new task naturally does not cause interference, while some completely overwrite retained information. Table 2a and Table 2b show the difference in accuracy during sequential learning (standard SNN without Nm) for two different permutations. We note that the second permutation (Table 2b) is harder than the first one, as the first two tasks were completely forgotten at step 3.

Table 2: Accuracies of two different permutations

(a) Permutation (1, 2, 3)

| Steps | Task 1 | Task 2 | Task 3 |
| --- | --- | --- | --- |
| 1 | 100% | . | . |
| 2 | 95.83% | 100% | . |
| 3 | 12.50% | 33.33% | 100% |

(b) Permutation (3, 1, 2)

| Steps | Task 3 | Task 1 | Task 2 |
| --- | --- | --- | --- |
| 1 | 100% | . | . |
| 2 | 33.33% | 100% | . |
| 3 | 4.17% | 0% | 100% |

To assess this, we count the total number of individual accuracies that dropped below a certain threshold throughout all permutations, stages of sequential learning, and random seeds, to capture the cases where catastrophic forgetting occurs (see Figure 3), since the mean accuracy only reflects the general performance and does not account for singular cases where the accuracy dropped sharply. In the setting of $\mathcal{D}_{test}(20)$ Figure 3a, we see that for Nm-SNN, no individual accuracy is below the threshold of 90%, and only 5% of accuracies are below 95%. On the other hand, the standard SNN clearly suffers from catastrophic forgetting; some classes are completely forgotten, 4% of accuracies are 0% and approximately 22% are below 40%. In the more difficult setting of $\mathcal{D}_{test}(50)$ Figure 3b, we observe heightened forgetting in the SNN; 15% of accuracies are 0%. While we see a decrease in performance for Nm-SNN compared to learning with 20 SGD updates, catastrophic forgetting is still mitigated as only 5% are below 50% and no task is completely forgotten.

Figure 4 shows the evolution of the accuracy of the first learned class with respect to every new SGD update on newer classes. At first, both the Nm-SNN and SNN accuracies are at 100%, and as soon as the 50 SGD updates of learning the second task have been completed (at the vertical dotted line) we can see that the SNN falls sharply to 40% while the Nm-SNN is at 90%. An interesting phenomena that we observe at the beginning of the third and final step of the sequential learning is the sudden increase in accuracy of the SNN on the first task even though it is learning the third task. An analysis of the spiking activity of the output layer of the SNN shows that during the first few batches, the output neuron corresponding to the current

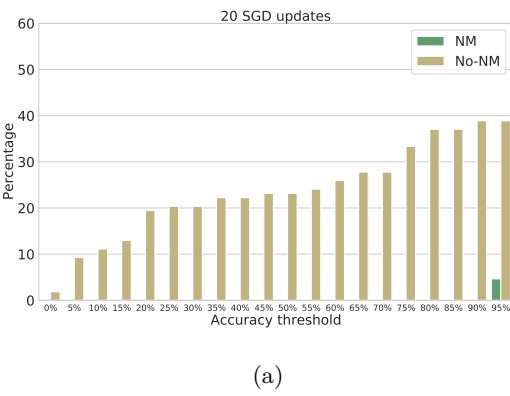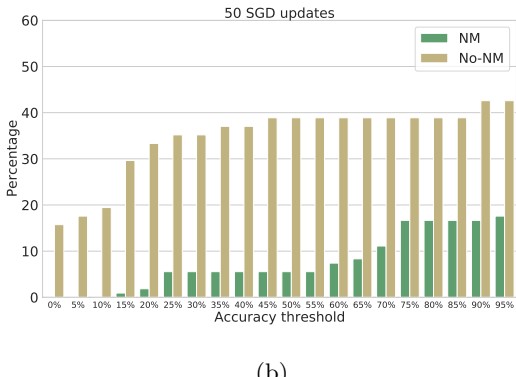

|        (a)        |        (b)        |

Figure 3: Percentage of individual accuracies ($y$-axis) that are below a certain threshold ($x$-axis) in DVS128 Gesture (lower is better). For example, in figure (b), the 4th bar from the left indicates that from the total number of accuracies that we evaluated during every permutation and step of CL, 30% of them dropped below 15% for the SNN and only 1% dropped below 15% for the Nm-SNN). The total number of accuracies is calculated for the $\mathcal{D}_{test}$ (20) Figure 3a and $\mathcal{D}_{test}$ (50) Figure 3b throughout all stages of sequential learning, all permutations, and multiple random seeds.

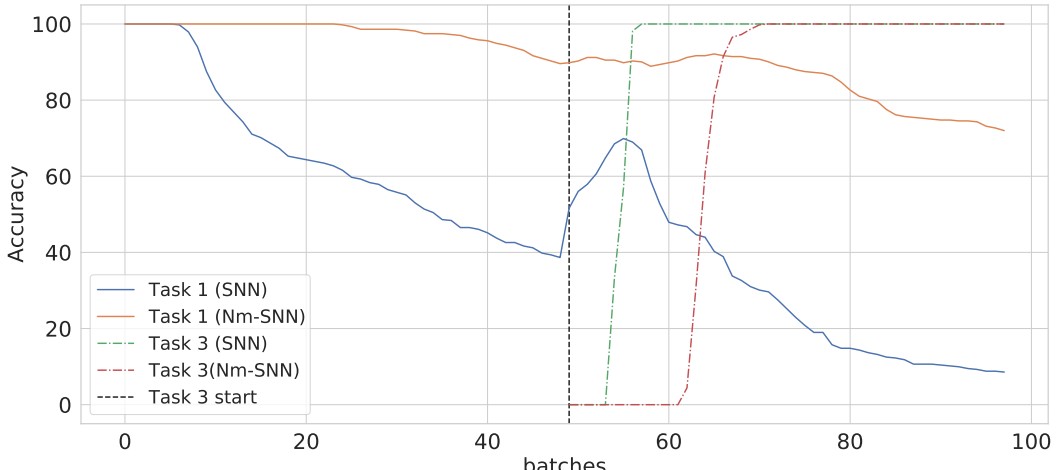

Figure 4: Evolution of the accuracy of the first learned class as new batches (SGD updates) of new classes are learned in the $\mathcal{D}_{test}(50)$ setting. The vertical dotted line represents the end of the second sequential task. Finally, the green and red lines represent the accuracy of the third and final learned task.

class does not spike at all and the accuracy is 0% at this stage (green dotted line). While learning task 3, the SGD updates tend to increase the synaptic weights of the SNN to encourage spiking. Due to the fact that some of these weights are also used for the first task, a spike in the output neuron corresponding to the first task occurs before, and we can see that once the accuracy of the third task increases (corresponding to a spike in the output neuron of the third task), the SNN starts forgetting the first task. However, even though the same phenomenon occurs for Nm-SNN, it is mitigated; we only observe a small increase in the first task and we also notice that learning the current task takes much longer (red dotted line). This is due to the fact that shared weights are being protected from the update by inhibiting the spiking of their pre-synaptic neurons through threshold modulation.

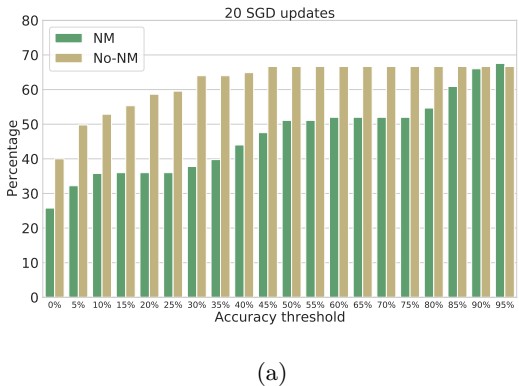
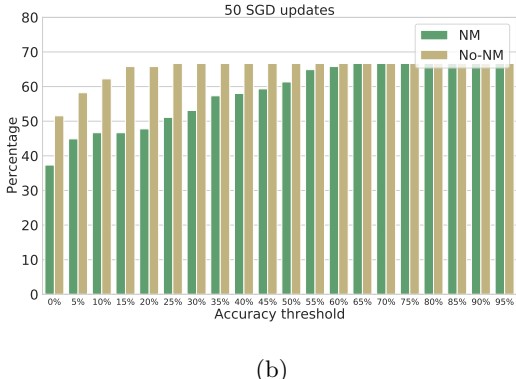

Figure 5: Percentage of individual accuracies ($y$-axis) that are below a certain threshold ($x$-axis) in EMNIST letters, the total number of accuracies is calculated for the $\mathcal{D}_{test}$ (20) Figure 5a and $\mathcal{D}_{test}$ (50) Figure 5b throughout all stages of sequential learning, all permutations and multiple random seeds.

## 4.2 EMNIST / MNIST

EMNIST letters is a more challenging test for our method since each class is composed of both uppercase and lowercase instances of letters, so the individual tasks are more difficult to learn and to retain. We also use more sequential tasks $n = 5$ for $\mathcal{D}_{evo}$ and $\mathcal{D}_{test}$ since we are not constrained with the total number of classes (26 letters) as in DVS128 Gesture. For the pre-training of the convolutional layers, we use the first 10 letters as classes noted by $\mathcal{D}_{pre}$. We ran the evolution for approximately 1000 generations using the next 5 letters. Each class is learned through 20 SGD updates with a batch size of 4 and 80 instances.

For the third test, we use a completely new dataset, MNIST, and double the number of sequential tasks to $n = 10$. We note this setting by $\mathcal{D}_{mnist,10}$.

Table 3: Mean accuracies for EMNIST/MNIST

| Settings | SNN (20) | EWC-SNN (20) | Nm-SNN (20) | SNN (50) | EWC-SNN (50) | Nm-SNN (50) |
|---|---|---|---|---|---|---|
| $\mathcal{D}_{evo}$ | 51.63% | 61.65% | **66.17%** | 48.92% | 52.46% | **60.25%** |
| $\mathcal{D}_{test}$ | 37.04% | 49.53% | **52.67%** | 34.31% | **49.12%** | 41.23% |
| $\mathcal{D}_{mnist,10}$ | 35.66% | **54.60%** | 44.23% | 32.13% | **42.95%** | 39.95% |

Even though the gap between SNN and Nm-SNN is smaller in EMNIST compared to DVS10 Gesture (see Table 3), especially in the hardest setting of $\mathcal{D}_{mnist,10}$ which tests the generalization to a completely new dataset, our method can effectively slow down forgetting in all settings in terms of mean accuracy. We note that EWC-SNN outperforms Nm-SNN on $\mathcal{D}_{mnist,10}$ in both regimes, and on $\mathcal{D}_{test}$ with 50 gradient steps. We interpret from this that EWC is able to adapt to the different task of digit analysis possibly through larger changes in the post-convolutional layer than was possible through surrogate gradient leading in the SNNs. We note that, as EWC is based on gradient updates and not activity modulation, advantages of the two methods could be combined.

Figure 5 also shows the improvement in retaining previous information between the two networks. However, we see that catastrophic forgetting still occurs in both cases (0% accuracies), a closer inspection shows that this occurs only in some permutations for Nm-SNN, while all permutations suffer from this problem for the case of SNN. It is possible that a better neuromodulator optimization protocol could solve this issue.

The comparison of the two distributions in Figure 6. indicates the improvement between paired accuracy samples, the t_test value for $\mathcal{D}_{test}$ (20) is 15.545 (with a p_value of $2.66 \times 10^{-15}$ ) and 14.98 (with a p_value of $2.43 \times 10^{-15}$ ) for $\mathcal{D}_{test}$ (50).

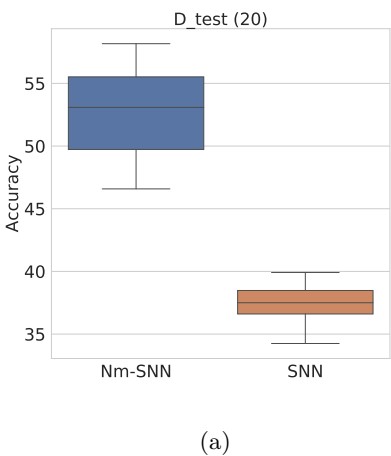
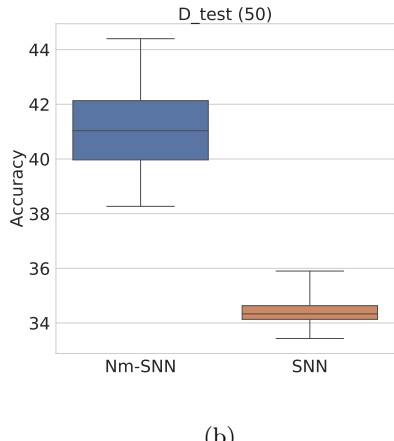

(a)

(b)

Figure 6: Distribution of accuracies of Nm-SNN (in blue) and SNN (in orange), for the settings of $\mathcal{D}_{test}$ (20) Figure 6a and $\mathcal{D}_{test}$ (50) Figure 6b for EMNIST letters

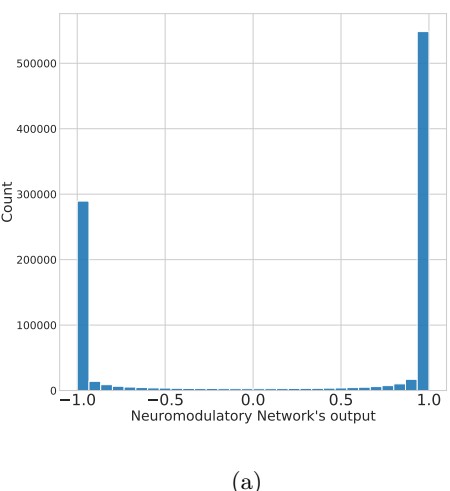
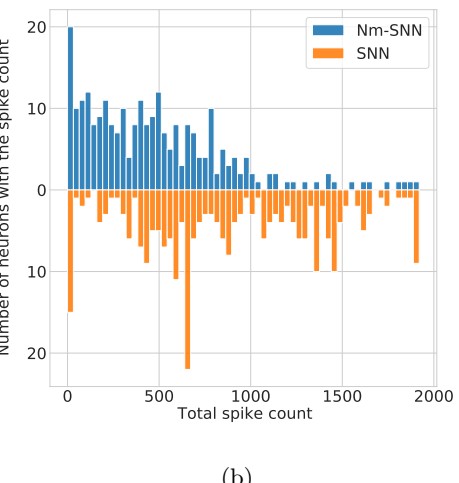

(a)

(b)

Figure 7: (a) represents the distribution of NmN outputs during a whole CL scenario; the output values are mainly grouped near $-1$ and 1. (b) represents the distribution of the total spike count of individual neurons; each bar represents the number of neurons that have a total spike count in the corresponding bin, for the Nm-SNN (blue upright) and SNN (orange upside down); for example, the first bar shows that there is 20 neurons in the Nm-SNN and 14 for the SNN that have a spike count between 0 and 30.

### 4.3   Analyzing Neuromodulatory Behavior

The motivation behind our work was that the NmN could potentially evolve to adapt the firing thresholds of neurons in such a way that enables a task-specific gating, so that the important neurons for previous tasks would not spike for new tasks. However, we did not use any constraints to enforce this behavior during the evolutionary optimization; the only criteria was the performance on the continual learning scenario. In this section, we analyze the behavior of an evolved NmN on the DVS128 Gesture setting (similar results were found on EMNIST/MNIST).

An evolved NmN's output converges to approximately binary threshold adaptation values of $\{-1, 1\}$ as shown in Figure 7a. During a complete CL scenario, about 63% of the NmN output values are positive,

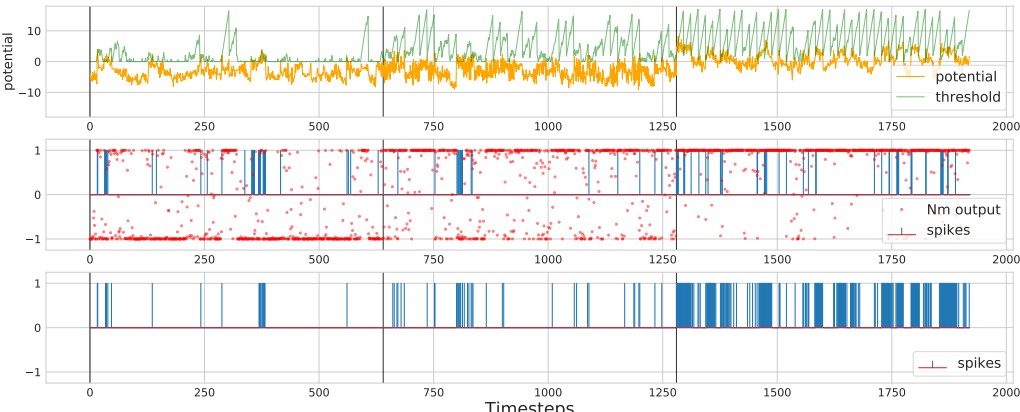

Figure 8: The characteristics of an individual neuron in the FC layer for every time-step of a full CL scenario, the vertical black lines separates between the three different sequential tasks. The first row represents the membrane potential and the firing threshold value, the second row represents the spikes (blue vertical bar) and the NmN output for the neuron's threshold adaption, and the third bar represents the spikes of the same neuron while not being neuromodulated.

which indicates that most outputs tend to restrain neurons from spiking which is consistent with the gating we expected. However, about 37% of outputs are negative, meaning that at some times neurons are encouraged to spike. Overall, if we compare the Nm-SNN to a standard SNN that's been through the same CL scenario, we find that the Nm-SNN has about 54% less spikes than the SNN. Thus, on the network level, the behavior of the NmN is more inhibitory.

To assess how the network's spiking activity is distributed between neurons of the neuromodulated FC layer, we can look at the distribution of the total spike count of individual neurons. Figure 7b shows this distribution for both Nm-SNN (in blue) and a standard SNN (in orange), and demonstrates that neuromodulation allows for a more regular distribution of spiking activity. The Nm-SNN's distribution shows that most neurons are concentrated on the lower total spike count region (less than 1000 spikes), while only few neurons spike more than 1000 times during the CL scenario. In comparison, the standard SNN's distribution doesn't show this regularity, with an important difference at the high total spike count regions. Without neuromodulation, there is a higher number of neurons that spike more than 1000 times. This difference could be explained by the outputs of the NmN, as the positive values restrain the most active neurons.

On the network level, the neuromodulator's behavior tends to reduce the overall spiking activity while evenly distributing it over the FC layer's neurons, these properties are suitable to mitigate catastrophic forgetting. If the spiking activity is concentrated in a few neurons, their corresponding weights would be updated each time they spike; a sparse and evenly distributed spiking activity could potentially allow for a separation of network parameters that are used for each task.

To analyze the NmN on a neuron level, we pick the same individual neuron and compare its characteristics while neuromodulated and while having a constant threshold. Figure 8 represents the membrane potential and threshold values of the neuron (1st row), its spikes while neuromodulated and its correspondent NmN output value with which its threshold was adapted during that time-step (2nd row) and finally the spikes of the neuron while not being neuromodulated (3rd row). The vertical lines in black separates between the three different sequential tasks in this CL scenario.

In the first row, the periodicity of the threshold value is due to the fact that after each input instance (16 time-steps) the threshold is reset to its base value of 1. The second row shows that the NmN outputs (red dots) are consistent with the previous network level analysis as they are mostly concentrated near the values $\{-1, 1\}$ with some exceptions that fall in between. Moreover, the figure shows that for this neuron, the NmN

exhibits a task-specific behavior; we can see that for the first task, most of the NmN outputs are negative thus encouraging the neuron to spike; while for the remaining two, most outputs are positive and restrain the neuron from spiking. This can also be seen by comparing the spike trains in the second and third row; we have 19% more spikes with neuromodulation for the first task and 23% and 88% less on the second and third tasks respectively. The behavior of the NmN on the final two tasks is what we were expecting: gating the activity can protect weights from being updated. This can be seen in particular for the third task where the extensive spiking activity of the neuron without neuromodulation leads to a lot of updates of its correspondent weights and ultimately to catastrophic forgetting. Furthermore, the NmN also encourages spiking in the first task. This shows that this neuron is mostly important for the first task and is restrained for the others. However, the behavior of the NmN is different for other neurons, where some are restrained on the first task and encouraged on others, while some are completely restrained on all tasks and so on.

While the overall behavior is task-specific, the NmN outputs in Figure 8 for the first and second task aren't exclusively negative or positive respectively: the NmN behaves differently for each input frame in every time-step. This could be explained by the fact that even though the input tasks are different, individual frames from the different tasks (for hand gestures for example) could have a resemblance, and thus the NmN treats them differently. This frame-specific behavior shows that using only the CL scenario performance as a metric during the evolutionary optimization can lead to richer and more flexible behaviors compared to imposing constraints or using handcrafted gating mechanisms.

## 5 Discussion

The study of SNNs and their properties could potentially lead to the development of faster and energy efficient neural networks that are less prone to the fundamental problems that plague standard ANNs. In this work, we demonstrated that threshold modulation could leverage an intrinsic property of SNNs to mitigate catastrophic forgetting in a continual learning setting where all network weights are shared and no task information is available, and where our neuromodulator has access only to local input, which is the previous convolutional layer's spiking activity. We used an evolutionary strategy to optimize our NmN where the criteria of selection is the performance on the continual learning scenario. This has lead to an NmN with a rich task-specific and frame-specific behavior; which is also generalizable to new tasks not seen during pre-training and evolutionary optimization.

This preliminary study on threshold modulation in SNNs has limitations which can be addressed through future work. The optimization of the NmN was performed with an evolutionary process, which could be replaced by more sample-efficient methods such as gradient descent. Further study of the optimized NmNs could also be used to develop generic threshold modulation policies which do not require further optimization.

Although our method was tested on simple networks and a small number of sequential taks, it opens the door for improved methods that leverage the intrinsic properties of SNNs, thus enabling the development of fast and energy efficient SNNs deployed on neuromorphic hardware that could learn continually in real world scenarios.

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
