# OpenReview forum: "Mitigating Catastrophic Forgetting in Spiking Neural Networks through Threshold Modulation"
_TMLR — Accepted by TMLR_

### Review · Reviewer_nxFu · 2022-06-27

**Summary Of Contributions:**

This paper proposed a neuromodulator network to dynamic control the threshold of spike neurons or another SNN for the catastrophic forgetting problem. The results show the neuromodulator network can significantly improve performance compared to SNN when learning the multiple tasks.

**Requested Changes:**

Suggestions:
1. In the background introduction, it would be better to briefly review existing methods of catastrophic forgetting problems on SNNs.
2. Insteading of using EMNIST, I think it's better to use the neuromorphic version of these image datasets like spike-based EMNIST, FMNIST, or N-MNIST.
3. As mentioned above, can the authors compare performance with existing papers?

Questions:
1. In the NM of figure 1, the outputs of neurons are passed to tanh function. Can the authors explain the motivation for adding this function?
2. What is the preprocessing step of DVS-128? Are methods like reducing dimension and shrinking time window used?

**Strengths And Weaknesses:**

Strength: I like the idea of using a new network to dynamically control the threshold. It seems to be another way to apply inhibition. The idea of using the evolutionary method rather than BP to optimize the neuromodulator network is also interesting. Overall, the methodology is good and the results are promising.

Weakness: This paper does not introduce the existing method for the catastrophic forgetting problems on SNN. It is hard to judge how good the proposed method is.

---

> ### Author Response · Authors · 2022-08-04
> **Response to reviewer nxFu**
>
> Thank you for the positive comments and constructive suggestions:
>
>
> ### Response to Suggestions:
>
> 1 - Following your suggestion, we added a paragraph on previous works that tackle the problem of catastrophic forgetting specifically for SNNs (in the section 2.3). We had originally not included these works as the networks they use are either trained in an unsupervised manner with STDP, or only not fully spiking (while having only the non-spiking layers as trainable). We have contextualized this difference in section 2.3.
>
> 2 - We do agree that neuromorphic datasets are fitting, which is why we studied DVS Gesture.  We also did experiments on N-MNIST but determined that it was redundant with the DVS128 experiment since both have similar total number of classes (n=10 and n=11); the results on N-MNIST were also similar. In order to test with a larger number of consecutive tasks we used EMNIST, and we are not aware that a neuromorphic version of this dataset exists.
>
> 3 - Please refer to the general response for the comparison with EWC.
>
> ### Response to Questions:
>
> 1 - We chose to pass the output of the neurons of the neuromodulator to the tanh function in order to keep the threshold adaptation at bounded values between [-1, 1] for every timestep (our base threshold value being 1) to avoid arbitrarily large threshold adaptations. Moreover, we also do not allow the threshold value to become zero or negative.
>
> 2 - For the preprocessing of the DVS128 dataset, we used the SpikingJelly framework which provides the pytorch dataset class. It does only event-to-frame integration, basically converting the raw event data to frames by summing the event values for each pixel for the frame duration. We used T=16 frames.

---

### Review · Reviewer_NavC · 2022-07-06

**Summary Of Contributions:**

The paper describes a method for continual learning. In an SNN architecture with two hidden layers, a separate neuromodulatory network is trained via evolutionary strategies to modulate the threshold of the second layer given the output of the first layer. The fitness function for the algorithm is the mean accuracy over a continual learning benchmark with permuted task order. In simulation on a DVS gesture and (E)MNIST datasets, this strategy reduces the impact of catastrophic forgetting.

**Broader Impact Concerns:**

No specific concern.

**Requested Changes:**

## Controls and baselines

- To give a better idea on the success of the method it would be great to compute the performance of control models. One simple control model would be to optimize the NmN network on the same loss function without evolutionary strategy but with gradient descent as for the rest of the SNN. This control makes sense to verify that it is not the architecture of the network which makes all the difference in performance.

- It would be great to implement at least one classical continual learning algorithm like EWC [ref. Kirkpatrick et al. 2017] or any other appropriate algorithm as a comparison.

- One thing that surprises me, fitness function requires many training step and seems rather slow. This is rather surprising because training a neural network with evolutionary strategy benefits from fast fitness evaluation. So I wonder whether there are any simpler version of a similar algorithm which would work better with faster fitness computation.

## Clarification

- Equation (9) and (10) $f'$ is used as if it was approximately equal to $\theta'$ but $f$ had a different definition which was given in equation (2) and (3). There seems to be some inconsistency here.

- Equation (9) and (10), I am surprised to see a pseudo derivative which is equal to the sigmoid function. Usually one has: $\theta'(x) \approx \sigma '(x) = \sigma(x) (1 - \sigma(x))$

- Unfortunately I could not read the performance in Figure 3 and 5. The sentence describing how to read these plots was a bit cryptic to me.

**Strengths And Weaknesses:**

## General opinion

The paper is well written and the training technique is quite clearly defined although there are a few inconsistencies.

The experiments are clearly described and the results seem to make sense but I find two major issues which makes the results incomparable with existing literature and hard to judge in terms of performance.

## Two major issues

(1) I do not understand what is the rationale for using spiking networks in combination with continual learning techniques.

My first problem is that there is no clear definition of what is a spiking neuron for the authors. If the motivation is to derive algorithm for neurmorphic hardware then it is typically assumed that all communications should be binary. However the tanh modulation requires communication from NmN to SNN which is not binary. If the choice of using spiking neurons is motivated by biology it is fine, but more explicit parallels with biological data would be needed to a more convincing model of biology.

It is not clear to me what is motivation to combine spiking neural networks with continual learning techniques. Since the authors rely on straight through estimators (called surrogate gradients in the context of spiking networks), the discontinuity are ignored and the spiking networks can be optimized as regular artificial neural networks. However I do not see why spiking networks would be more or less affected by catastrophic forgetting in comparison with non-spiking artificial neural networks. In this sense, I think that having spikes in this paper does not make a crucial technical difference and the method could have been compared to other methods which are not necessarily applied to spiking networks.

(2) This lead me to the second major issue: there are no comparison with other continual learning techniques on the presented benchmarks.

Unfortunately, the method is not compared to other continual learning techniques despite the existing literature which is well reviewed and cited at the beginning of the paper. Since the task is custom for this paper I cannot judge whether the performance is very good although it looks like it was done seriously.

Also comparisons with other control algorithm using the same implementation could have been performed to provide more data points to judge the quality of the algorithm. I provide some suggestions in the following section of the review.

---

> ### Author Response · Authors · 2022-08-05
> **Response to reviewer NavC**
>
> Thank you, we appreciate your feedback and have integrated it. We think especially the comparison to classical continual learning methods was an important addition to our paper.
>
> ## Responses to issues:
>
> To respond to the first issue raised on motivation: we believe that the firing threshold mechanism of SNNs can be harnessed as a natural control mechanism to avoid catastrophic forgetting.
>
> Concerning Surrogate Gradient Learning, the discontinuity is only ignored in the backward pass. However, the main difference and motivation for using SNNs for continual learning lies in the forward pass; if a presynaptic neuron does not spike (membrane potential < threshold), all of its corresponding weights will have a gradient update of 0. This is how the forward pass controls the backward updates.
>
> In this way threshold modulation can increase the threshold of neurons to block them from spiking and protect their weights, or reduce the threshold to encourage spiking and learning.
> Other methods that we cited use similar gating mechanisms on ANNs (by using masks or other techniques). In our work, we try to leverage the threshold mechanism as a natural property of SNNs to gate activity and reduce catastrophic forgetting.
>
> Concerning a possible application to neuromorphic hardware, our goal was to study how a neuromodulatory network could mitigate catastrophic forgetting only by adapting the threshold. While this is not directly application to neuromorphic hardware due to the tanh output activation function of the NmN, our analysis does show that the outputs of the NmN converge to mostly discrete values {-1, 1}, hinting at the possibility of discretizing our model for implementation needs.
>
> We believe that the results demonstrate the motivation for studying continual learning in SNNs; threshold modulation can reduce catastrophic forgetting as the firing mechanism acts as a natural activity gate and allows for selective gradient updates. This is not a default property of ANNs and requires activity masking such as Dropout.
>
> ## Response to Controls and Baseline:
>
> 1 - We thank the reviewer for this idea and did perform this control experiment. Since the loss of the SNN is only calculated on the current task, optimizing the NmN for this loss with backpropagation helps with learning the current task (similar to adding extra parameters to the SNN) and aggravates forgetting. For easier tasks, we found no difference between doing this and using the standard SNN, however for more complex tasks, this helps the SNN to converge faster and therefore forget faster. As the evolutionary loss is calculated only at the end on the aggregate behavior of the NmN, it is less prone to this effect. We did not include this control in the modified version of the article but could include it in an appendix if requested.
>
> 2 - 	Please refer to the general response for the comparison with EWC.
>
> 3 - We thank the reviewer for this suggestion, which we did test. We found that it was important to include multiple permutations in the fitness function to encourage generalization to different task orderings. However, we did try a simplification to the fitness function by evaluating the previous tasks only at the end of the scenario instead of doing it every step and the gain in computation time was considerable with only about -0.5% loss in accuracy. We did not rerun all experiments with this simplification but we will include it in the open-source code.
>
> ## Response to Clarifications:
>
> 1, 2 - We corrected both equation (9) and equation (10), there was an inconsistency in using the symbol f, and an error in the sigmoid derivative.
>
>
> 3 - We do agree that the Figure 4 and 5 were not clearly explained, we added more explanations to address this. The mean accuracy over all the different permutations and CL steps does not capture the sharp drops in accuracy that may occur on a single CL scenario. We counted every time an accuracy from all those that we evaluate drops below a certain threshold, in order to know the number of times an accuracy of any task dropped below 50%, for example. The motivation behind this is to evaluate where accuracy drops significantly over the tasks and CL steps.

---

> > ### Comment · Reviewer_NavC · 2022-08-12
> > **Conclusion**
> >
> > Thank you the taking my suggestions seriously and updating the paper accordingly.
> >
> > I do not have any specific request for improvements or modifications at the moment.
> >
> > Thank you for adding the comparison with EWC. Unfortunately, I cannot judge whether the implementation of EWC is faithful and the performance is satisfactory on this task which was designed specifically for this paper.
> >
> > Overall, my opinion remains unchanged: I do not find that there is a technical innovation regarding the way spiking networks are used (it follows the usual surrogate gradient framework) and dealing with spiking neurons does not make any difference with regards to the continual learning algorithm. The efficiency of the algorithm and the performance could have been described with non-spiking networks and -- very probably -- it would not have made any difference. Instead with a more classical non-spiking demonstration, it would have been accessible to a broader community and it would have made performance comparisons with other algorithms easier. The use of spikes is particularly questionable for this paper because of the quasi-absence of concrete links with spike-restricted neuromorphic hardware or models of biological neurons.

---

### Review · Reviewer_yN44 · 2022-07-21

**Summary Of Contributions:**

This paper proposes a voltage threshold modulation method to mitigate the catastrophic forgetting problem in SNN during continual learning. The reviewer thinks the paper has three major contributions:

1. The authors observed the gating effect of the voltage threshold on weight update when training SNN using backprop. By changing the threshold, spiking neurons can have different activation behaviors. This directly influences the weight update magnitude for synapses to the post-synaptic neurons.

2. The paper introduced a separate spiking network (NmN) trained by an evolutionary algorithm to control the voltage threshold of neurons in the last fully-connected layer of the SNN. By modulating the thresholds, the network regulates the update of output synaptic weights in the continual learning scenario.

3. The authors performed experiments on evet-based and static datasets. The proposed method generates better performance than a regular SNN and prevents accuracy drop on old classes when a new class is learned.

**Broader Impact Concerns:**

There is no concern about the ethical implications of the work.

**Requested Changes:**

Important Changes:

1. The authors need to include clear descriptions of partitions used in the dataset. From the manuscript, we know D_evo is used to perform evolutionary optimization (EA). However, the results in Table 1 and Table 3 also included D_evo for continual learning (CL). Is D_evo divided into EA and CL datasets? If so, how much data is used for evolutionary optimization and generating the results in the tables? Since the regular SNN doesn't need evolutionary optimization, is the SNN use the same group of data in D_evo for CL?

2. In addition, how does the experiment divide the data of each class when performing continual learning? How much data is used to train the new class and test the trained model? Since the D_evo has much fewer data per class for continual learning than D_test, how does the experiment deal with this difference when combining them for D_6 in Table 1?

3. The authors need to clarify experiment details related to training and testing the Nm-SNN during continual learning. Is the number of neurons in the output layer fixed and equal to the number of classes? Or new output neurons are added when new classes are introduced in the continual learning? Is the NmN used only during training the SNN? Or both in training and testing?

4. Are the hyperparameters for the regular SNN optimized for best performance? We know hyperparameters like voltage thresholds can significantly influence the performance of SNN. The authors need to show they have optimized the performance of the SNN in the tasks.

5. Since a large amount of the NmN output is -1, as shown in Figure 7(a), will the network prevent the voltage threshold go below zero? If not, how will the neuron behave when the voltage threshold is negative?

6. The authors give an overclaim that the proposed method performs better for dynamic datasets in the last paragraph of Section 4.3. The accuracy numbers between the DVS-Gesture and the EMNIST datasets are not comparable. These two datasets have different sizes and numbers of classes. The difference in accuracies doesn't give enough evidence to support the claim in the paper.

Other Changes:

1. The authors need to add detailed explanations on the relationship between voltage threshold and weight update, preferably using math expressions. This information is needed to complete the background of the proposed method and for readers who are not that familiar with the concept.

2. Figures 3 and 5 show the accuracy differences between Nm-SNN and the regular SNN. However, it's unclear from the figure how the accuracy is evolved in the time dimension when new tasks are learned. Adding an additional time dimension in both figures will be better to show how the accuracies change after learning new tasks.

3. The quality of Figure 7 is low. First, the word "count" is not complete in Figure 7(a). Second, the number of neurons for SNN should be positive on the y-axis of Figure 7(b).

**Strengths And Weaknesses:**

Strength

1. Gating the weight update via controlling the voltage threshold is natural for the SNN. The proposed method doesn't require any additional changes to the spiking neuron model and has the potential to be deployed on neuromorphic processors.

2. The proposed method doesn't require any class-specific information when changing the class to be trained. This makes it has the potential to be used in real-world tasks when the task or class transition is not as transparent as the artificial dataset used in the paper.

3. Experiments in the paper demonstrate the effectiveness of the proposed method in mitigating the catastrophic forgetting problem during continual learning. The Nm-SNN achieved much higher accuracy than the regular SNN in both DVS-Gesture and EMNIST datasets. In addition, Figure 4 of the paper shows the Nm-SNN forgets much slower than the regular SNN.

4. The analysis of the neuromodulatory behavior within the Nm-SNN (Section 4.3) shows changing the voltage threshold during learning can effectively prevent the classification overfit to a specific feature. By solving this overfitting problem in the output layer, the proposed method can prevent aggressive weight updates when learning new classes. From analyzing the results in Figure 8, the reviewer thinks this is what truly happens inside the Nm-SNN and how it can mitigate the catastrophic forgetting problem.

Weakness

1. The evolutionary strategy used to optimize the NmN is very costly. The algorithm needs to iterate all tasks in the evolution dataset for every individual group of randomly generated parameters. This is not generalizable for large datasets and large NmN. First, increasing the number of tasks and data for optimization will directly increase the iteration times. Second, increasing the size of the NmN for more complex tasks will exponentially increase the search space for the evolution algorithm.

2. The backprop-based approach to optimize the NmN may not be as memory costly as the paper claims. Although the training needs to iterate all tasks and data in the dataset, intermediate gradient summations can be used to reduce the memory cost. Unlike the evolution algorithm, the gradient-based method may give a more reliable optimization direction requiring fewer update epochs.

3. The proposed method only solved the catastrophic forgetting problem in a linear classifier. Since all parameters except the output layer are fixed during continual learning, what the experiment truly trained is a linear classifier using the features generated by the pre-trained SNN. However, learning on a linear classifier has limited ability on complex tasks. This may explain why the SNN has poor performance on the EMNIST.

4. Building upon the previous point, the proposed method may not be able to work for training parameters in a multi-layered network. In the current setting, the parameters in the output layer for different classes have no interference. During training, forgetting may happen when the data of two classes share similar features. For example, the feature represented by the neuron in Figure 8 of the paper is shared for all three classes. Thus, overstimulation in one class may trigger forgetting in other learned classes. However, a multi-layered network will have a different mechanism for forgetting during continual learning. There will be much more interference in the parameters for different classes. Thus, good results on a linear classifier cannot guarantee the method works on a multi-layered network.

---

> ### Author Response · Authors · 2022-08-05
> **Response to reviewer yN44 (1/2)**
>
> We thank you for your extensive review, we appreciate your comments and reasoning, the summary of contributions and strength summaries perfectly our work. We respond to individual comments below:
>
> ## Response to Weaknesses:
>
> 1 - We do agree that scaling the NmN optimization would be costly. In this preliminary study, we show that threshold modulation is an effective means of reducing catastrophic forgetting. Determining a policy that modulates thresholds could be achieved through other means besides the evolution used in this work, such as gradient descent. Also, further study of the NmN could be used to propose general threshold modulation policies which don’t require optimization. We have now mentioned these limitations and possible future directions in section 5.
>
> 2 - We agree that training an NmN with gradient descent instead of evolution is a worthwhile direction for future research. We have modified section 3.3 to present this more clearly and are interested in using gradient descent in the future. We believe that the current method and results are still of interest, as the main conclusion concerns the capacity of threshold modulation to deter catastrophic forgetting and not the optimization of the NmN itself. While the suggestion of intermediate gradient calculations is appreciated, implementing and testing this compared to the current NmN would be a considerable modification and we are unsure of the impact this would have on the performance in the CL scenarios.
>
> 3, 4 - Thank you for this detailed reasoning. We believe there was confusion concerning the final layer: due to the spiking activation of the final neurons, the final layer is non-linear. We have modified Figure 1 to replace the word “Linear” with “Dense” in the diagram as this was meant to reflect the weight type, not the activation type.
> We do agree that a linear classifier has limited capacity, and even with the spiking activation, freezing all but the final layer does negatively impact performance on EMNIST. We therefore also tried an application of the neuromodulator when both the hidden layer and the final output layer are trained; we achieved similar results and plan on adding this experiment into the appendix. We find that this experiment, along with the current results, demonstrates that our method works on a non-linear model and when multiple layers are being trained.
> We did not test on architectures which have more than two hidden dense layers, as most convolutional architectures only use two final layers. We believe that using an architecture with multiple dense hidden layers would benefit from threshold modulation on each layer, but that these policies would mostly likely need to be different. This could be a direction for future work, but as mentioned above we believe the current method demonstrates that neuromodulation is influencing the training of more than a linear classifier.
>
> ## Response to Requested changes:
>
> ### Important Changes :
>
> 1, 2  - We added multiple modifications to the paper to try and clarify these points, the additions are:
> - For $\mathcal{D}_{evo}$, we use different instances for EA and for the tests (results on table 1). Except for the setting of 50 gradient step where the small size of the dataset doesn’t allow us to do so, in this setting we reuse part of the instances used in evolution.
> - For all datasets, either the instances used for EA or tests, we divide them into 80% used for training and 20% used to evaluate the accuracy.
> - Finally, for D_6 in the DVS128 gesture section, since we don't have enough classes to test for a bigger number of consecutive tasks. we used  D_evo (3 classes) and D_test (3 classes) to make D_6.
> If D_evo = {a, b, c}  and D_test  = {e, f, g}    (classes IDs)
> Then D_6 = {a, e, b, f, c, g}
>
> 3 - For the first point, yes the number of neurons in the output layer is equal to the number of classes in our dataset (even though we don’t do it but it is possible to extend it by adding new output neurons, our method is compatible with this), we mention this in section 3.2 Neuromodulated SNN, in the end of the second paragraph.
> For the second point, the NmN is used both during training and testing of the SNN, we added this clarification in section 3.2.
>
> 4 - Hyperparameters for SNNs like base voltage thresholds, membrane time constant (for LIF neurons) among others could be optimized to significantly improve performance. However, we didn’t perform any hyperparameter optimization for the SNN itself and used standard values for the DVS Gesture dataset from the SpikingJelly library.
>
>
> 5 - This is an important point that we didn’t explain initially, we added to the paper (subsection 3.2 Neuromodulated SNN) how we treat this:
> “We don't allow threshold values to be zero or negative (for biological plausibility purposes), thus the final threshold adaptation is $max(0 , V_{th} [t] + \Delta V_{th} [t]) + \epsilon$

---

> ### Author Response · Authors · 2022-08-05
> **Response to reviewer yN44 (2/2)**
>
> 6 - We have removed this claim; our intent was to point out the difference in the two datasets and how this could be related to the difference in neuromodulatory behavior, but we agree that there is not enough analysis presented to support this claim.
>
> ### Other Changes:
>
> 1 - We respectfully believe that the existing surrogate gradient literature provides this relationship in detail, specifically the article Neftci et al., 2019, which we cite.
> Emre O. Neftci, Hesham Mostafa, and Friedemann Zenke. Surrogate gradient learning in spiking neural networks: Bringing the power of gradient-based optimization to spiking neural networks. IEEE Signal Processing Magazine, 36(6):51–63, 2019. doi: 10.1109/MSP.2019.2931595
> The relationship between weight update and voltage threshold used in our work is the same as that established in surrogate gradient (see page 55 and equations 8-11 in Neftci et al); the only modification we propose is that this threshold can be modified. We use a constant surrogate function for weight updates. We do note that the gradient update is 0 when there is no spike, which is influenced by threshold modulation.
>
> 2 - Figures 3 and 5 weren’t explained clearly, also suggested by another reviewer, we added some more explanations to make it clearer. These figures show how many accuracies from those we evaluate dropped below a certain threshold throughout all the permutations and steps of CL scenarios. On the other hand, Figure 4 shows the evolution of accuracy in the time dimension.
>
> 3 - Thank you for the remark, we corrected both Figure 7(a) and 7(b).

---

### Author Response · Authors · 2022-08-05
**General response to reviewers**

We thank all the reviewers for their helpful reviews; we appreciate the positive comments and constructive feedback from which we added multiple modifications to the paper (highlighted in red). We clarified multiple points which we hope aids in understanding our work.

The largest substantive change to the article comes from a request from two reviewers to compare to existing methods. We therefore added a comparison with EWC, which is a widely used baseline for many continual learning papers, in Tables 1 and 3. We discuss these results in section 4. We do believe it is important to note that our method and EWC do not follow the same experimental protocol. EWC requires task specific information: whenever a change has occurred in the task to be trained, EWC needs to perform computations on the previous task to approximate the Fisher information term. The neuromorphic method proposed does not require information about the change in tasks and is online. We have now explained that difference in section 4.

We respond individually to each suggestion in detail below

---

### Decision · Action_Editors · 2022-09-06

**Recommendation:** Accept with minor revision

**Comment:**

Given the evaluations of the reviewers, I can propose acceptance of the manuscript, given that some minor revisions are made.
The reviewers raised the following point:
- The proposed method to mitigate catastrophic forgetting was restricted to the final layer, potentially limiting the performance on some data sets and the scalability of the method. The authors extended the analysis to hidden layers in the revision, but the results did not significantly improve. Hence, the claim that the approach could be scaled to larger networks and more complex problems is not well supported by the shown evidence.
The authors should discuss this point in the Discussion and adjust their claim on scalability made there accordingly.